# Viral zoonoses assessment in invasive rodent species from São Tomé and Príncipe

Tereza Almeida[1,2], Maria Carolina Matos[1,2], Daniel A. Velarde-Garcéz[1,2], Vanessa A. Mata[1,2], Marquinha Martins[3], Conceição Neves[3], Pedro Beja[1,2,4], Ana M. Lopes[1,2,5,6], Joana Abrantes[1,2,7] *

1 CIBIO, Centro de Investigação em Biodiversidade e Recursos Genéticos, InBIO Laboratório Associado, Campus de Vairão, Universidade do Porto, Vairão, Portugal, 2 BIOPOLIS Program in Genomics, Biodiversity and Land Planning, CIBIO, Campus de Vairão, Vairão, Portugal, 3 Birdlife International, São Tomé and Príncipe Office, Bairro do Hospital, Água-Grande, São Tomé, São Tomé e Príncipe, 4 CIBIO, Centro de Investigação em Biodiversidade e Recursos Genéticos, InBIO Laboratório Associado, Instituto de Agronomia, Universidade de Lisboa, Lisboa, Portugal, 5 UMIB—Unit for Multidisciplinary Research in Biomedicine, ICBAS—School of Medicine and Biomedical Sciences, University of Porto, Porto, Portugal, 6 ITR, Laboratory for Integrative and Translational Research in Population Health, Porto, Portugal, 7 Departamento de Biologia, Faculdade de Ciências, Universidade do Porto, Porto, Portugal

* jabrantes@cibio.up.pt

## Abstract

Zoonoses are diseases transmitted from animals to humans, highlighting the inseparable link between animal and human health. They are responsible for approximately 2.4 billion cases of illness and 2.2 million deaths annually, posing a significant challenge to public health and food security. Transmission of infectious agents from animals to humans occurs through direct contact, ingestion, inhalation, or inoculation of the infectious agent. Finding practical ways to monitor the presence and/or abundance of zoonotic pathogens is important to estimate the risk of spillover to humans. Since rodents are present almost everywhere, live in proximity with humans and host several zoonotic viruses, we conducted a screening in different tissue samples of black rats (*Rattus rattus*) and brown rats (*Rattus norvegicus*) collected in São Tomé and Príncipe in 2021 and 2022 for the presence of five zoonotic families of viruses, including *Arenaviridae*, *Coronaviridae*, *Flaviviridae*, *Hantaviridae*, and *Poxviridae*. Although we found no evidence of the presence of these viral taxa among the rodent samples tested, our study does not exclude their presence in São Tomé and Príncipe. Continued monitoring of these and other zoonotic viruses is advisable to prevent or mitigate the emergence of viral diseases that are often fatal to humans.

## Introduction

The archipelago of São Tomé and Príncipe is located on the equator in the Gulf of Guinea in West Central Africa. It includes two main islands, São Tomé and Príncipe, and several islets. The archipelago has a relatively small area of around 1000 km²

**Data availability statement:** All relevant data are within the manuscript and its Supporting information files.

**Funding:** The study was supported by BirdLife International and the European Union through the project "Action for Sustainable Landscape Management in São Tomé and Príncipe" (ENV/2020/420-182). It received extra support via the European Union's Horizon 2020 research and innovation program under grant agreement No. 854248. The work was further co-funded by the project TROPIBIO NORTE-01-0145-FEDER-000046 and NORTE-01-0246-FEDER-000063, supported by Norte Portugal Regional Operational Programme (NORTE2020), under the PORTUGAL 2020 Partnership Agreement, through the European Regional Development Fund (ERDF). Fundação para a Ciência e a Tecnologia (FCT) funded the individual research contracts of VAM and AML (CEECIND/02547/2020, https://doi.org/10.54499/2020.02547.CEECIND/CP1601/CP1649/CT0004; CEECIND/01388/2017, https://doi.org/10.54499/CEECIND/01388/2017/CP1423/CT0007 and the PhD scholarships of MCM and DVG (2023.02359.BD and 2023.02833.BD, respectively). The funders played no role in the study design, data collection and analysis, decision to publish, or preparation of the manuscript.

**Competing interests:** The authors have declared that no competing interests exist.

and is ca. 250 km off the mainland. São Tomé and Príncipe have a unique biological diversity, with a high number of endemic species of birds, amphibians, and plants, as a result of its orography, climate and geography [1].

Discovery and colonization of the archipelago in the 1470s led to the introduction of several exotic species, such as wild pig (*Sus scrofa*), mona monkey (*Cercopithecus mona*), African civet (*Civettictis civetta*), weasel (*Mustela nivalis*), house mouse (*Mus musculus*), black rat (*Rattus rattus*) and brown rat (*Rattus norvegicus*). Other domestic and feral mammals, such as dogs, cats, pigs, goats, cows, horses, and others, have recently colonized the archipelago. Thus, all non-flying mammals present nowadays in the archipelago of São Tomé and Príncipe, except the endemic shrews *Crocidura thomensis* and *Crocidura fingui*, were introduced [1,2].

The introduction of invasive species had a detrimental impact on the native flora and fauna of São Tomé and Príncipe, being associated with habitat degradation and loss, competition for food and space, and predation. Indeed, the predation of adults, juveniles and nests of some critically endangered species by rats, monkeys, civets, and weasels has been highlighted by BirdLife International [3]. Rats and civets have successfully colonized primary forest areas, including river edges [3], and exert a significant impact by consuming eggs and preying on juveniles. Additionally, introduced rodents can serve as reservoirs for infectious diseases such as leptospirosis, plague, toxoplasmosis, hantavirus and arenavirus infections, amplifying their impact [4].

The black rat is among the top 100 worst invasive species globally due to its negative impact on the ecosystems [5]. Numerous studies detailed their significant impact on natural ecosystems, especially on islands [3,6–8]. Furthermore, they have wide-ranging implications for human activities and health, including agriculture and infrastructure damage, spread of diseases, contamination of food supplies, and increased public health costs [4,9–11]. By living in close contact with humans and domestic animals, having an almost worldwide distribution (except Antarctica and some islands) [12], they are important reservoirs of zoonotic viruses [11,13–17]. Rodents are primary mammalian reservoirs for zoonotic pathogens due to the high pathogen diversity they harbour, causing around 90 diseases, including >30 viral zoonoses alongside bacterial, helminthic, protozoan, and fungal infections [9,10].

While humans host many viruses without clinical signs, several rodent-borne viruses can cause severe, and often fatal, diseases in humans, e.g., lymphocytic choriomeningitis virus (LCMV), Lassa virus (LASV), severe acute respiratory syndrome coronavirus (SARS-CoV), Middle East respiratory syndrome coronavirus (MERS-CoV), tick-borne encephalitis virus (TBEV), dengue virus, Hantaan virus (HTNV), Seoul virus (SEOV), Puumala virus, monkeypox (MPXV) and cowpox (CPXV) viruses, among others [11,13–17]. Humans can be infected either through rodent biting, contact with water, food, and surfaces contaminated with infected urine and/or feces, inhalation of aerosols, or via arthropod vectors [18].

This study aimed to detect and assess the prevalence of zoonotic viruses in invasive black and brown rats from the archipelago of São Tomé and Príncipe as human and non-human communities are more vulnerable to diseases introduced by invasive species. This vulnerability stems from ecological, biological, and social factors [19].

We focused on selected viral taxa with zoonotic potential previously detected in rodents from other African countries, namely *Poxviridae*, *Arenaviridae*, *Coronaviridae*, *Flaviviridae*, and *Hantaviridae* [20–25]. *Arenaviridae* and *Hantaviridae*, which include Lassa fever and hantavirus pulmonary syndrome-causing viruses, pose serious public health risks [e.g., 22,25]. *Coronaviridae*, including SARS-CoV-2, *Flaviviridae*, such as yellow fever and Zika viruses, and *Poxviridae* like cowpox, have been reported in African rodents, highlighting their role in zoonotic transmission [20,21,23,24].

## Materials and methods

### 1. Specimen collection

Sampling was carried out in two field trips: one pilot trip and one extended field season. The pilot trip took place in December 2021, during which several trapping techniques were explored in cocoa plantations, palm plantations, and forests. The second field trip occurred from May to September 2022 across 12 localities evenly spread among the following habitats: forests, cocoa and palm plantations, and villages (Fig 1). These habitats represent a gradient of human activity, with sampling reflecting varying degrees of contact intensity between rats and humans. Rodents were captured using Tomahawk-like traps baited with oat cookies and peanut butter. Sampling in each locality lasted 4 days, in a 1-hectare plot containing 36 traps, each spaced 20 meters apart. Traps were set up on the morning of the first day and checked daily after dawn and before dusk. The species, sex, and body measurements of each captured animal were recorded (see S1 Table for individual data). Animals were anesthetized with isoflurane and then euthanized by cervical dislocation; all efforts were made to minimize suffering. Liver, spleen and fecal samples were collected and stored in RNA*later*® (Thermo Fisher Scientific) at room temperature until the end of the day, after which they were transferred to a freezer. Samples were

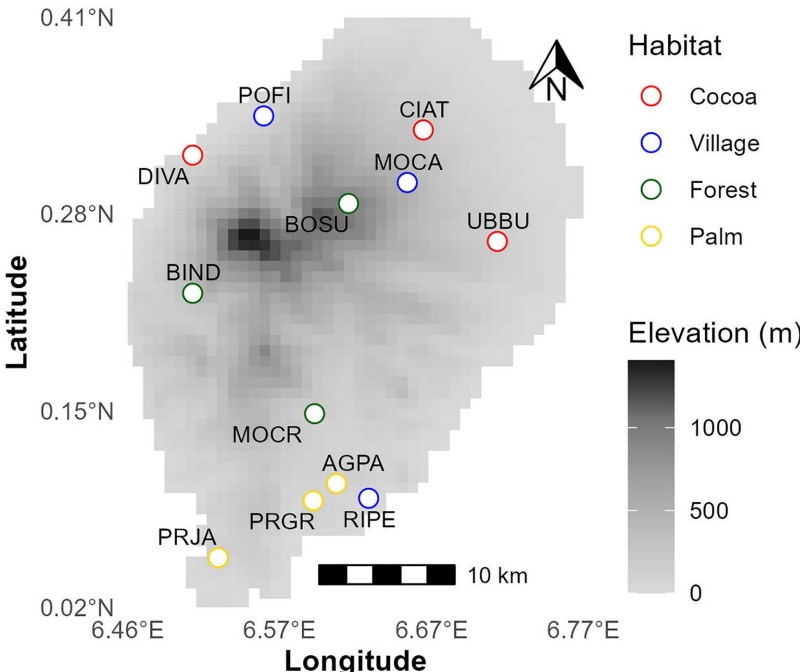

**Fig 1. Map of the São Tomé island with the location of the different habitats where the samples were collected.** Waypoints correspond to field-collected sampling sites. The habitats appear with different color codes: cocoa (red), villages (blue), forest (green) and palm (yellow). Digital elevation model (DEM) data were obtained from the GMTED2010 dataset (U.S. Geological Survey, public domain) at 7.5 arc-second resolution, from earthexplorer.usgs.gov, cropped to the island polygon. DEM and waypoint data are visualized in latitude/longitude (WGS84). Elevation values are in meters above sea level and are represented by a color gradient from white to black.

shipped to the laboratories of CIBIO-InBIO, Vairão, Portugal. Collection and shipment of the samples were done according to local legislation. Sampling was approved by "Direção das Florestas e da Biodiversidade" of São Tomé and Príncipe, and shipping was certified by "Centro de Investigação Agronómica e Tecnológica de São Tomé e Príncipe (CIAT-STP)". The protocol was approved by the Animal Welfare and Ethics Review Body of CIBIO-InBIO (ORBEA-BIOPOLIS-CIBIO).

## 2. DNA and RNA extraction

DNA and RNA were extracted from 10–30 mg of liver (n = 81), spleen (n = 80) and feces (n = 80) of *R. rattus* (n = 70) and *R. norvegicus* (n = 15) using the EasySpin Genomic DNA Tissue Kit SP-DT-250 (Citomed) and the GeneJET RNA Purification kit (Thermo Fisher Scientific), respectively, according to the manufacturer's protocol. cDNA was synthesized with the NZY first-strand cDNA synthesis kit (NZYtech), according to the provided protocol.

## 3. DNA viruses screening

Detection of poxviruses with zoonotic potential was performed by conventional PCR using pan-pox universal primers available in the literature (Table 1). Reactions were carried out with 1 µL of DNA in a final volume of 10 µL containing 5 µL of Phusion Flash High-Fidelity PCR Master Mix (Thermo Fisher Scientific) and 2 pmol of each oligonucleotide. Conditions for PCR amplification were: 98°C for 10 sec, 45 cycles of 98°C for 1 sec, 48°C for 5 sec and 72°C for 4 sec, and a final extension at 72°C for 1 min. A myxoma virus (*Leporipoxvirus*) positive sample available at CIBIO-InBIO was included as a positive control.

## 4. RNA viruses screening

Detection of RNA viruses was performed by conventional RT-PCR using pan/universal primers available in the literature (Table 2). Reactions were carried out with 1 µL of cDNA in a final volume of 10 µL containing 5 µL of Phusion Flash High-Fidelity PCR Master Mix (Thermo Fisher Scientific) or Supreme NZYTaq II 2x Green Master Mix (NZYtech) and 2 pmol of each oligonucleotide. For each virus family, a positive sample was included as control of the reaction: for *Coronaviridae*, SARS-CoV-2 (kindly provided by Dr. Susana Lopes, CIBIO-InBIO); for *Flaviviridae*, Bagaza virus (kindly provided by Dr. João Queirós, CIBIO-InBIO); for *Arenaviridae*, lymphocytic choriomeningitis virus (LCMV; pLCMV_Pol-I-L-ARM; provided by the University of Basel, Department of Biomedicine, as participant of the European Virus Archive); for *Hantaviridae*, Hantaan virus (HTNV 78–118; provided by the Biomedical Research Center of the Slovak Academy of Sciences, as participant of the European Virus Archive).

For coronaviruses, conditions for PCR amplification were: 98°C for 5 sec, 40 cycles of 98°C for 1 sec, 51°C for 5 sec and 72°C for 2 sec, and a final extension at 72°C for 1 min. The PCR product was used as a template for a second PCR to improve virus detection, using the same conditions.

For flaviviruses, conditions for PCR amplification were: 95°C for 5 min, 35 cycles of 94°C for 30 sec, 57°C for 25 sec and 72°C for 5 sec, and a final extension at 72°C for 10 min.

For the arenavirus detection, conditions for PCR amplification were: 98°C for 10 sec, 30 cycles of 98°C for 1 sec, 52°C for 5 sec and 72°C for 7 sec, and a final extension at 72°C for 1 min.

For the hantavirus detection, a nested PCR was performed that targets a conserved region of the L gene [25]. The first PCR conditions were: 98°C for 10 sec, 7 cycles of 98°C for 1 sec, 50°C for 5 sec and 72°C for 15 sec, 35 cycles of 98°C

**Table 1. List of primers and PCR conditions used for detection of DNA viruses.**

| Virus family | Primer ID | Sequence 5'-3' | Amplicon size (bp) | Reference |
|---|---|---|---|---|
| *Poxviridae* | PanPox Low-GC F | ACACCAAAAACTCATATAACTTCT | 220 | [26] |
| | PanPox Low-GC R | CCTATTTTACTCCTTAGTAAATGAT | | |

**Table 2. List of primers and PCR conditions used for detection of RNA viruses.**

| Virus family | Primer ID | Sequence 5'-3' | Amplicon size (bp) | Reference |
|---|---|---|---|---|
| *Arenaviridae* | LVL3359A-plus | AGAATTAGTGAAAGGGAGAGCAATTC | 394 | [27] |
| | LVL3359D-plus | AGAATCAGTGAAAGGGAAAGCAATTC | | |
| | LVL3359G-plus | AGAATTAGTGAAAGGGAGAGTAACTC | | |
| | LVL3754A-minus | CACATCATTGGTCCCCATTTACTATGATC | | |
| | LVL3754D-minus | CACATCATTGGTCCCCATTTACTGTGATC | | |
| *Coronaviridae* | PanCoV-11Fwd | TGATGATGSNGTTGTNTGYTAYAA | 154 | [28] |
| | PanCoV-13Rev | GCATWGTRTGYTGNGARCARAATTC | | |
| *Flaviviridae* | PFlav-F | TACAACATGATGGGAAAGAGAGAGAARAA | 270 | [29] |
| | PFlav-R | GTGTCCCAKCCRGCTGTGTCATC | | |
| *Hantaviridae* | HAN-L F1 | ATGTAYGTBAGTGCWGATGC | 412 | [25] |
| | HAN-L-R1 | AACCADTCWGTYCCRTCATC | | |
| | HAN-L- F2 | TGCWGATGCHACIAARTGGTC | | |
| | HAN-L-R2 | GCRTCRTCWGARTGRTGDGCAA | | |

for 1 sec, 55°C for 5 sec and 72°C for 15 sec and a final extension at 72°C for 1 min. The second PCR conditions were: 98°C for 10 sec, 40 cycles of 98°C for 1 sec, 60ºC for 5 sec and 72°C for 15 sec, and a final extension at 72°C for 1 min.

## Results and discussion

Rodents exhibit remarkable adaptability and thrive in environments closely intertwined with humans and anthropogenic activities, fostering the emergence of rodent-borne zoonotic viruses [19]. Monitoring of rodents has been proposed as an effective tool for surveilling zoonotic diseases, since they host a large plethora of viruses and play a key role in the transmission of infectious agents. In this study, we collected a total of 241 samples from 70 black rats (*R. rattus*) and 15 brown rats (*R. norvegicus*). Samples were collected in different habitats, broadly following an anthropogenic gradient, from villages to palm and cocoa plantations, and to forests, covering a wide spectrum of contact intensity between rats and humans. The samples included liver (n = 81), spleen (n = 80), and feces (n = 80) with an average of 20 samples (range 16–26) per species per habitat (Table 3). Variation in sample size among species reflects differences in abundance and habitat use. Indeed, since the sampling design aimed to balance the effort across land-use types, i.e., forest, cacao, palm, and village, rather than balance the number of samples per species, brown rats, which are rare outside human settlements, are represented by fewer samples. Overall, the values for the morphological parameters were within the range of those reported for these species [https://www.gbif.org/species/2439270; https://www.gbif.org/species/2439261].

All samples were screened using conventional PCR and RT-PCR to detect the presence of DNA or RNA from viral taxa with zoonotic potential. Selection of the taxa was based on their previous identification in rodents from other African countries, specifically *Poxviridae*, *Coronaviridae*, *Flaviviridae*, *Arenaviridae,* and *Hantaviridae* [20–25]. During our surveillance, no positive results were obtained in the black and brown rats for the five viral taxa tested.

DNA virus screening was directed towards poxviruses, as human cases of cowpox virus were reported with an origin in infected pet rats [30]. Nonetheless, poxviruses were rarely reported in rodents. This is in line with our results, where no poxviruses have been identified in brown and black rats, and also with other previous surveys in Europe [31,32]. In other rodent species, orthopoxviruses have been detected (including the brown rat, gerbil, common vole, and Western Mediterranean mouse [20,21,27]).

A meta-analysis on the prevalence of orthohantaviruses in rodents (~5%; [33]) highlighted that their global prevalence is a point of concern. The role of rodents as reservoirs prompted our analysis. There is no molecular evidence of hantavirus in brown and black rats from São Tomé and Príncipe. Its prevalence in rodents worldwide is variable, ranging from 0%

**Table 3. Morphological parameters (mean±S.D.) of male and female samples of black rats (*Rattus rattus*; A) and brown rats (*Rattus norvegicus*; B) per habitat.**

**A)** *Rattus rattus*

| Habitat | Sex | Body weight (g) | Body length (mm) | Tail length (mm) | Hind foot length (mm) | *n* |
|---------|-----|-----------------|------------------|------------------|-----------------------|-----|
| Forest | Male | 175.3±39.2 | 193.3±19.8 | 210.4±20.2 | 25.9±0.9 | 10 |
| | Female | 182.2±15.3 | 193.2±9.0 | 220.2±15.7 | 25.2±0.7 | 5 |
| Cacao | Male | 176.6±54.9 | 186.3±36.8 | 201.8±20.6 | 25.9±1.9 | 8 |
| | Female | 123.3±35.4 | 168.4±28.6 | 188.2±25.8 | 25.3±4.2 | 10 |
| Palm | Male | 164.6±46.5 | 184.5±19.4 | 204.3±1.2 | 25.5±1.2 | 11 |
| | Female | 173.4±38.2 | 190.9±17.8 | 207.1±17.3 | 24.6±1.5 | 10 |
| Village | Male | 153.5±31.4 | 191.2±19.7 | 205.7±14.8 | 24.4±0.8 | 6 |
| | Female | 130.0±13.2 | 185.8±8.5 | 205.0+16.3 | 24.2±0.9 | 10 |

**B)** *Rattus norvegicus*

| Habitat | Sex | Body weight (g) | Body length (mm) | Tail length (mm) | Hind foot length (mm) | *n* |
|---------|-----|-----------------|------------------|------------------|-----------------------|-----|
| Forest | Male | 195 | 215 | 214 | 26.6 | 1 |
| | Female | – | – | – | – | – |
| Cacao | Male | – | – | – | – | – |
| | Female | 206 | 216 | 164 | 26.6 | 1 |
| Palm | Male | – | – | – | – | – |
| | Female | 179.7±36.8 | 206.3±12.1 | 173.7±10.1 | 24.3+1.7 | 3 |
| Village | Male | 176.3±76.2 | 202.0±26.1 | 158.3±24.0 | 27.1±1.1 | 6 |
| | Female | 196.3±22.9 | 205.7±18.2 | 158.7+12.7 | 27.3±1.9 | 3 |
| | Undetermined | 222 | 210 | 170 | 26.4 | 1 |

to almost 30% (e.g., [27,32,34,35]). To what concerns the African continent, the orthohantavirus Anjo strain was found in the black rat in Madagascar [36].

Flaviviruses, which encompass viruses such as the yellow fever virus, have been detected in rodents from Senegal, Uganda and Kenya [24,37], including black rats. Current global changes, such as increased trading and traveling, have altered the distribution, transmission cycles and dynamics of flaviviruses, highlighting the critical need for ongoing surveillance studies. Our survey did not detect any flavivirus in black or brown rats, consistent with other studies [38]. On the contrary, antibodies reactive with flaviviruses have been found in black rats and house mice in Mexico [39].

For all viral taxa included, our results could not determine if the tested animals had been exposed to the targeted viruses, but rather if they were currently infected. For example, in Europe, a seroprevalence of 3.6–11.7% against LCMV (arenavirus) was found [40–42], while for Puumala virus and Dobrava virus (hantaviruses) it stays below 0.5% [40]. Thus, serological testing combined with molecular testing could have provided more information regarding the zoonotic risk associated with these species.

While no coronavirus RNA has been found in our dataset, their remarkable capacity to overcome species barriers and adapt to hosts close to humans emphasizes the critical need for comprehensive research on coronavirus infections [e.g., 43]. Other efforts to survey coronavirus presence in rodents have yielded conflicting results. While in Europe and in the neighboring country Gabon no coronaviruses have been found in rodent species [31,32], in the Republic of Congo two animals were found to be positive for coronavirus RNA (the Congo forest mouse, *Deomys ferrugineus*, and the big-eared swamp rat, *Malacomys longipes*). Moreover, the ability of SARS-CoV-2 variants of concern to spread has been demonstrated [44] along with evidence of urban rats' exposure to the virus, due to the presence of anti-SARS-CoV-2 antibodies. Collectively, these results underscore the critical need to monitor potential reservoir species in insular environments to effectively control pathogen spread.

Our results were negative for the presence of arenaviruses, consistent with the findings of a study conducted in Italy [32]. However, this group of viruses is known for causing persistent, silent infections in rodents and severe, and even lethal, human diseases [45]. Lassa virus is endemic in West Africa and black rats were shown to be competent reservoirs [46]. Its prevalence among black rats is especially high in some regions of Nigeria, reaching >75% prevalence [22]. While Lassa fever cases have not yet been reported in São Tomé and Príncipe, viral presence in neighboring countries, such as Nigeria, Benin, and Togo, increases the risk of arrival to the islands. Interestingly, in West African countries, where the invasive black rat competes with the native Natal multimammate mouse (*Mastomys natalensis*) and primary reservoir host of Lassa virus, the zoonotic spillover risk is lower when in the presence of the black rat [47]. Moreover, a significant diversity of mammarenaviruses was discovered within a limited host sample size and a relatively small geographical area, reinforcing that Angola is a hotspot for arenaviruses. These viruses have been identified in species such as *Mastomys natalensis*, *Micaelamys namaquensis* and *Mus triton* that are in close contact with other rodent species and humans [48]. Based on current distribution records and available literature, there is no confirmed evidence of these species occurring on the island of São Tomé [49].

In our survey, we tested representative samples (liver, spleen, and feces) of all individuals that covered the broad tissue and cell tropism of the selected viruses. Samples were collected at different locations, seasons, and years, which ensured unbiased sampling of males and females, adults and juveniles. The detection techniques were appropriate and of high quality, with negative and positive controls confidently confirming their applicability. Yet, positive controls corresponded to samples with active infections and high viral loads, and the samples from this study were collected in apparently healthy animals, with no conspicuous clinical signs that could indicate disease and/or an ongoing active infection. Hence, target viruses might be present but in very low viral loads that may not have generated viremia, going unnoticed in PCR. Some of these viruses do not cause lifelong persistent infections, and sampling may not coincide with the space-time presence of infected hosts, reducing the likelihood of viral detection. Several other limitations might explain the results obtained. Indeed, the low number of samples analyzed from only a few sites and covering short periods of time might have precluded the detection of active and acute infections. Further surveys should aim at a larger sample size to increase the likelihood of detection. Other limitations include sample viability (despite their storage in RNA later, they were not tested on-site), low sensitivity of the PCR or the limited specificity of the primers employed, notwithstanding their pan/universal application, considering that the targeted viruses might have a particular evolution in this insular region. To overcome these limitations, more sensitive methodologies might be considered as methods of choice for future surveillance studies, such as high-throughput sequencing (HTS) or digital PCR. Indeed, HTS allows unbiased detection of all microorganisms in a sample and has been successfully employed to identify uncommon and novel infectious viral agents [reviewed in 50], while digital PCR is a highly sensitive and precise molecular technique that detects minimal amounts of the target even in the presence of a high number of non-target templates [51]. The combination of continued surveillance and correct management of invasive species is expected to result in significant improvements in both public and animal health and the conservation of biodiversity.

These apparently negative results should be taken with caution, as they do not exclude a potential infection state among *R. rattus* and *R. norvegicus* from São Tomé and Príncipe, being advisable that their handling should be safely conducted due to the extensively documented risk they pose as hosts for zoonotic viruses. The publication of absence data on rat-borne pathogens provides a more complete and unbiased assessment of zoonotic risk. Even if these are true negative results, they remain valuable for constructing distribution models and are essential to identifying potential areas at risk.

Here, we show the results of a molecular survey investigating zoonotic viruses in invasive rodent species. Our study highlighted an absence of such viruses among rats from São Tomé and Príncipe, making, to our knowledge, the first survey of its kind conducted in this insular country. Rodents are widely distributed in urban, peri-urban, and rural environments, where their increasing proximity of mammals to humans and livestock amplifies the vulnerability of native species.

In line with a One Health approach, we propose maintaining long-term surveillance of zoonotic viruses in São Tomé and Príncipe. This surveillance effort should be complemented with serological studies and extended to include other viral zoonoses of concern (e.g., rabies, Ebola virus, and highly pathogenic avian influenza) and other species. Furthermore, viruses with limited zoonotic potential but known to infect humans – such as rotavirus, norovirus, astrovirus, and kobuvirus – should also be investigated, particularly those sporadically detected in wild rats and mice.

## Supporting information

**S1 Table. Species, sex, and body measurements of the individuals sampled in the different habitats: cacao (A), village (B), forest (C) and palm (D).**
(PDF)

## Acknowledgments

Positive control samples were kindly provided by Dr. Susana Lopes, CIBIO-InBIO (*Coronaviridae*: SARS-CoV-2); Dr. João Queirós, CIBIO-InBIO (*Flaviviridae*: Bagaza virus); Department of Biomedicine, University of Basel, as participant of the European Virus Archive (*Arenaviridae*, Lymphocytic choriomeningitis virus: pLCMV_Pol-I-L-ARM); Biomedical Research Center of the SLovak Academy of Sciences, as participant of the European Virus Archive (*Hantaviridae*, Hantaan virus: HTNV 78-118).

We want to acknowledge the Direção das Florestas e da Biodiversidade (DFB) of São Tomé and Príncipe for approving the collection of samples by local legislation, and the Centro de Investigação Agronómica e Tecnológica de São Tomé e Príncipe (CIAT-STP) for certifying the shipment of samples.

## Author contributions

**Conceptualization:** Ana M. Lopes, Joana Abrantes.

**Funding acquisition:** Marquinha Martins, Conceição Neves, Pedro Beja, Joana Abrantes.

**Investigation:** Tereza Almeida, Maria Carolina Matos.

**Methodology:** Tereza Almeida, Maria Carolina Matos, Daniel A. Velarde-Garcéz, Vanessa A. Mata, Marquinha Martins, Conceição Neves.

**Project administration:** Joana Abrantes.

**Supervision:** Ana M Lopes, Joana Abrantes.

**Writing – original draft:** Tereza Almeida.

**Writing – review & editing:** Tereza Almeida, Maria Carolina Matos, Daniel A. Velarde-Garcéz, Vanessa A. Mata, Marquinha Martins, Conceição Neves, Pedro Beja, Ana M. Lopes, Joana Abrantes.

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
