## [Decision Letter · Decision Letter 0]

20 Oct 2025

PLOS ONE

Dear Dr. Abrantes,

Thank you for submitting your manuscript to PLOS ONE. After careful consideration, we feel that it has merit but does not fully meet PLOS ONE’s publication criteria as it currently stands. Therefore, we invite you to submit a revised version of the manuscript that addresses the points raised during the review process.

We look forward to receiving your revised manuscript.

Kind regards,

Pablo Colunga-Salas

Academic Editor

PLOS ONE

**Journal Requirements:**

4. Please include a new copy of Table 1 in your manuscript; the current table is difficult to read. Please follow the link for more information: https://journals.plos.org/plosone/s/tables

**Additional Editor Comments:**

Your contribution is considered valuable to the field of infectious diseases, particularly in a country where available information remains limited. However, in accordance with the reviewers’ recommendations, I kindly request that you address each of their comments in order for the editorial process to continue.

Reviewers' comments:

Reviewer's Responses to Questions

**Comments to the Author**

1. Is the manuscript technically sound, and do the data support the conclusions?

Reviewer #1: Yes

Reviewer #2: Yes

2. Has the statistical analysis been performed appropriately and rigorously?

Reviewer #1: N/A

Reviewer #2: N/A

3. Have the authors made all data underlying the findings in their manuscript fully available?

Reviewer #1: Yes

Reviewer #2: Yes

4. Is the manuscript presented in an intelligible fashion and written in standard English?

Reviewer #1: Yes

Reviewer #2: Yes

Reviewer #1: The article Viral zoonoses assessment in invasive rodent species from Sao Tomé and Principe is well-written; however, it does not constitute a significant contribution to the topic of viral zoonoses carried by two invasive rat species. While the subject is interesting—addressing invasive rat species on an island, where literature has reported their highest risk to native wildlife—the authors also note that these rats have been present in the area for a long time and, in fact, only coexist with two species of wild mammals (two shrews). From my perspective, the article is not suitable for publication in this journal, which seeks higher-impact articles, at least not in its current form. Below, I provide a series of suggestions to consider for improving the text.

The sample size used for the analyses (86 specimens) is very small, and although the authors themselves discuss this limitation, it would be desirable to obtain a larger sample, particularly for the brown rat (R. norvegicus). Even if this species is not very abundant on the island, the current sample (15 or 14 specimens) remains insufficient. This may be due to the limited number of traps employed, prompting my recommendation to increase their use in future efforts.

In the Materials and Methods section, it is not specified that sampling followed a gradient, an aspect that is mentioned later in the text.

Although the Materials and Methods section states that specimens were weighed, measured, and sexed, these data are not discussed further. It would be useful to include this information at least in a table to provide an overview of the collected population's condition. Additionally, it is mentioned that specimens were collected along an anthropogenic gradient, but the number of specimens per land cover type is not specified. I suggest summarizing this information in a table as well. Even though no specimens tested positive, data on their distribution across the landscape would enrich the text.

I recommend conducting additional tests, possibly for antibody presence, which, given the current results, might provide further insights into the presence of these viruses in rat populations.

Finally, I suggest enhancing the text with a map of the study area to provide context for international readers unfamiliar with the region.

I have left additional comments within the text.

Reviewer #2: Despite having negative results, the research is well-structured in terms of results and discussions. However, it is necessary to explain how the sample size was determined and why the N varies from one species to another.

**Do you want your identity to be public for this peer review?** For information about this choice, including consent withdrawal, please see our Privacy Policy

Reviewer #1: No

Reviewer #2: No

---

## [Author Response · Author response to Decision Letter 1]

13 Nov 2025

Reviewer #1: The article Viral zoonoses assessment in invasive rodent species from Sao Tomé and Principe is well-written; however, it does not constitute a significant contribution to the topic of viral zoonoses carried by two invasive rat species. While the subject is interesting—addressing invasive rat species on an island, where literature has reported their highest risk to native wildlife—the authors also note that these rats have been present in the area for a long time and, in fact, only coexist with two species of wild mammals (two shrews). From my perspective, the article is not suitable for publication in this journal, which seeks higher-impact articles, at least not in its current form. Below, I provide a series of suggestions to consider for improving the text.

- We are grateful for the reviewer’s constructive feedback and fully acknowledge the concerns expressed. The present study used samples originally collected as part of a separate project investigating the dietary preferences of species inhabiting São Tomé and Príncipe. As outlined in the manuscript, the study’s limitations reflect the logistical challenges of the field sample collection, but are also due to financial constraints. Despite these limitations, we consider it valuable to report negative results to reduce knowledge gaps and avoid redundant research efforts.

The sample size used for the analyses (86 specimens) is very small, and although the authors themselves discuss this limitation, it would be desirable to obtain a larger sample, particularly for the brown rat (R. norvegicus). Even if this species is not very abundant on the island, the current sample (15 or 14 specimens) remains insufficient. This may be due to the limited number of traps employed, prompting my recommendation to increase their use in future efforts.

- Sampling was designed to cover a gradient of human activity that would reflect a range of rodent-human contact intensities and aimed at an average of 20 samples per species per habitat. The sampling design aimed to balance the effort across land-use types, i.e., forest, cacao, palm, and village, rather than balance the number of samples per species. While black rats (R. rattus) are abundant in the four sampled habitats and appear well-represented, brown rats (R. norvegicus) are rare outside human settlements and are represented by fewer samples. Thus, the variation in sample size among and between species reflects species differences in habitat use and abundance. Regarding the sampling effort, we used Tomahawk-like traps baited with oat cookies and peanut butter and placed 20 meters apart in a 1-hectare plot, and sampling lasted 4 days. These settings are in agreement with those used in other studies of these species.

As mentioned in the manuscript, we conducted two field trips, one pilot trip in December 2021 to test for several trapping techniques, and a second field trip from May to September 2022 across 12 localities evenly spread among the different habitats (cocoa, palm, forest and villages), where the most successful trapping techniques were employed. While not mentioned in the manuscript, no R. norvegicus were successfully trapped in the pilot field trip, hinting for their difficult sampling, which was further confirmed in the second field trip. However, as we were able to collect a few individuals in the second field trip, we opted to include them in the analyses for a more complete overview of the zoonotic potential of rodents in São Tomé and Príncipe.

In the Materials and Methods section, it is not specified that sampling followed a gradient, an aspect that is mentioned later in the text.

- This information was included in the Materials and Methods section: “These habitats represent a gradient of human activity, with sampling reflecting varying degrees of contact intensity between rats and humans.”

Although the Materials and Methods section states that specimens were weighed, measured, and sexed, these data are not discussed further. It would be useful to include this information at least in a table to provide an overview of the collected population's condition. Additionally, it is mentioned that specimens were collected along an anthropogenic gradient, but the number of specimens per land cover type is not specified. I suggest summarizing this information in a table as well. Even though no specimens tested positive, data on their distribution across the landscape would enrich the text.

- As suggested, a new table summarizing the information on morphological parameters (mean ± S.D.) of male and female samples of black rats and brown rats collected per habitat was added to the manuscript (c.f. Table 3). The values for the morphological parameters are within the range of those reported for these species (www.gbif.org). Moreover, a table with individual data on those parameters was also included as supplementary information (c.f. S1 Table).

I recommend conducting additional tests, possibly for antibody presence, which, given the current results, might provide further insights into the presence of these viruses in rat populations.

- We fully agree with the reviewer’s recommendation; however, the current project’s financial framework preclude the inclusion of additional tests in the current study.

Finally, I suggest enhancing the text with a map of the study area to provide context for international readers unfamiliar with the region.

- A map of the study area was included in the manuscript (Figure 1).

I have left additional comments within the text.

- All additional comments within the text have been carefully addressed, including corrections to formatting and clarification of sample differences noted between the Materials and Methods and Results and Discussion sections.

Reviewer #2: Despite having negative results, the research is well-structured in terms of results and discussions. However, it is necessary to explain how the sample size was determined and why the N varies from one species to another.

- Sampling was designed to cover a gradient of human activity that would reflect a range of rodent-human contact intensities, and aimed at collecting an average of 20 samples per species per habitat. The sampling design intended to balance the effort across land-use types, i.e., forest, cacao, palm, and village, rather than balance the number of samples per species. Thus, since black rats (R. rattus) are abundant in the four sampled habitats, they appear well-represented. In contrast, brown rats (R. norvegicus) are rare outside human settlements and are represented by fewer samples. The variation in sample size among and between species reflects differences in habitat use and abundance.

---

## [Editor Report · Decision Letter 1]

11 Jan 2026

Viral zoonoses assessment in invasive rodent species from São Tomé and Príncipe

PONE-D-25-32689R1

Dear Dr. Abrantes,

We’re pleased to inform you that your manuscript has been judged scientifically suitable for publication and will be formally accepted for publication once it meets all outstanding technical requirements.

Kind regards,

Pablo Colunga-Salas

Academic Editor

PLOS One
---

## [Editor Report · Acceptance letter]

PONE-D-25-32689R1

PLOS One

Dear Dr. Abrantes,

I'm pleased to inform you that your manuscript has been deemed suitable for publication in PLOS One. Congratulations! Your manuscript is now being handed over to our production team.

Kind regards,

on behalf of

Pablo Colunga-Salas

Academic Editor

PLOS One